# Multi-Agent Temporal Task Solving and Plan Optimization

**Primary Keywords:** *(6) Temporal Planning; (7) Multi-Agent Planning;*

## Abstract

Several multi-agent techniques are utilized to reduce the complexity of classical planning tasks, however, their applicability to temporal planning domains is a currently open line of study in the field of Automated Planning.

In this paper, we present MA-LAMA, a centralized, unthreaded, satisfying, total-order, multi-agent temporal planner, that exploits the 'multi-agent nature' of temporal domains to, as its predecessor LAMA, perform plan optimization.

In MA-LAMA, temporal tasks are translated to the constrained *snap-actions* paradigm, and an automatic agent decomposition, goal assignment and required cooperation analysis are carried to build independent search steps, called Search Phases. These Search Phases are then solved by consecutive agent local searches, using classical heuristics and *temporal constraints*.

Experimentation shows that MA-LAMA is able to solve a wide range of classical and temporal multi-agent domains, performing significantly better in plan quality than other state-of-the-art temporal planners.

## Introduction

In numerous scenarios the systems we plan for are naturally viewed as multi-agent (MA) systems, and, additionally, MA systems inherently tend to require some degree of concurrency between agents to operate efficiently. Some examples of temporal MA planning (MAP) scenarios that present 'MA nature' can be found in the International Planning Competition (IPC) benchmark domains, e.g. Rovers and Satellites, with independent homogeneous agents; and Elevators or Logistics, where interaction between agents is required.

In contrast, Automated Planning literature has classically considered temporal and MAP as two individual lines of work. This can be viewed in the MAP Survey (Torreño et al. 2017), where it is stated that the handling of scenarios derived from temporal MA systems is an open point that needs to be addressed, as MAP solvers tend to focus on classical planning domains.

In that survey, they also present some classifications for MAP solvers in terms of their taxonomy: *threaded*, with interleaved planning and coordination, and *unthreaded*, dealing with planning and coordination separately. Their computational structure: *centralized*, with a monolithic design and a central process, and *distributed*, sharing the planning task

across multiple processing units. And their privacy preservation: providing *strong*, *object cardinality* or *weak* privacy, depending on the extent to which each agent's sensitive information is preserved. For plan quality optimization (one of the main points of interest of this work), MAP systems tend to make use of cost aware classical planners, like LAMA (Richter and Westphal 2010).

MAP techniques used across all types of MAP solvers are different from those used in temporal planners, which tend to revolve around time and numeric reasoning to deal with the inherent complexity of concurrent actions search spaces (Rintanen 2007). Our objective with this paper is to test the MAP techniques' effectiveness in dealing with temporal complexity, especially in areas where time reasoning-based planners struggle, as not coupled with makespan plan quality optimization.

MA-LAMA is the result of the integration of these techniques: a centralized, unthreaded, satisfying, total-order, temporal MAP system that can deal with MA temporal tasks. Our contribution is centered around the application of automated task decomposition, goal assignment, and required cooperation MAP techniques to temporal tasks. This results in a planning algorithm that exploits the MA and concurrent nature of temporal MA domains.

The paper is structured as follows. The next section presents the related literature. Then MA-LAMA is presented, first broadly and after, in a detailed per-component view. Finally, we provide an empirical evaluation of the planner, followed by the conclusions and future work.

## Related Work

Literature on both MAP and temporal planning is extensive as separate lines of work. To our knowledge, TFPOP (Brafman and Domshlak 2008) is the only MA planner that is able to deal both with time and durative-actions. It follows a centralized scheme, producing non-linear plans that maintain a thread of sequentially ordered actions per agent, exploiting the concept of *coordination points* for loosely coupled agents and through CSP+planning. However, it was not compared to any other MAP solver.

**Regarding MA planning**, some examples of threaded MAP planners are partial-order-planning (POP) based planners, as MH-FMAP (Torreño, Sapena, and Onaindia 2015), which computes distributed versions of $h_{DTG}$ and $h_{Land}$

heuristics; MAD-$A^*$ (Nissim and Brafman 2013), an optimal solver based in local agent evaluation of each state; GPPP (Maliah, Shani, and Stern 2014), which builds a relaxed public plan before each local planning stage; and MAPlan (Fišer, Štolba, and Komenda 2015), which follows a flexible distributed implementation of local agent state search methods, as well as both local and global heuristics, as $h_{FF}$ and *LM-Cut*, showing strong performance specially with distributed computation agents.

For unthreaded MAP planners, some approaches to this type are: PMR (Luis, Fernández, and Borrajo 2020), based in plan merging and reuse with simultaneous planning by all agents; CMAP (Borrajo and Fernández 2015), with weak privacy preservation in agents assembly and single-agent search; Distoplan (Fabre et al. 2010), an optimal planner that exploits independence between agents, not limiting their possible interactions beforehand; PSM (Tožička, Jakubuv, and Komenda 2015), which expands Distoplan and introduces Planning State Machines: agent local task representations that can be merged or projected; $A^\#$ (Jezequel and Fabre 2012), with cost informed and constrained factored planning following $A^*$ search; and DPP (Maliah, Shani, and Stern 2016), one of the best performing unthreaded MAP solvers through accurate public projection of MAP task information with object cardinality privacy preserving.

Other MA techniques include symmetry score based task decomposition for classical (Nissim, Apsel, and Brafman 2012) and numeric planning (Shleyfman, Kuroiwa, and Beck 2023).

In our case, MA-LAMA deals with MAP solving with a traditional approach, considering sequential total-order planning. We also make use of several MAP techniques not yet mentioned, such as: task decomposition into local search regions (Lansky 1991), exploiting loosely coupled agents from TFPOP, *coordination points* detection and constraints definition from Planning First (Nissim, Brafman, and Domshlak 2010), distributed planning graphs with coordination constraints from DPGM (Pellier 2010), required cooperation (Zhang, Sreedharan, and Kambhampati 2016), also used in the MARC planner (Sreedharan, Zhang, and Kambhampati 2015); satisfiability through sequential MAP task solving from $\mu$-SATPLAN (Dimopoulos, Hashmi, and Moraitis 2012), and automatic MAP agent decomposition from ADP (Crosby, Rovatsos, and Petrick 2013).

Several of these MAP planners, and others, participated in the 2015 Competition of Distributed and Multi-Agent Planning (CoDMAP) (Komenda, Štolba, and Kovács 2016), being the top performers ADP in coverage and CMAP-q in quality for the centralized track, and PSM and MAPlan for the distributed track overall.

**Regarding temporal planning**, several temporal planners incorporate techniques that MA-LAMA makes us of, such us the *snap-actions* paradigm, continuous numeric effects treatment and temporal frontier state constraints from POPF (Coles et al. 2021) and OPTIC (Benton, Coles, and Coles 2012), both from the COLIN (Coles et al. 2012) family of planners; and TFLAP (Sapena, Marzal, and Onaindia 2018) multi-heuristic search based in $h_{FF}$ and $h_{Land}$, which had good performance in the 2018 IPC temporal track.

Other participants in this competition were POPCORN (Savaş et al. 2016), which is able to operate with control parameters, TemPorAl (Cenamor et al. 2018), a portfolio that was the top performer in the competition, and CP4TP (Furelos-Blanco and Jonsson 2018), another portfolio. From 2014 IPC temporal track, notable participants were: IT-SAT (Feyzbakhsh Rankooh and Ghassem-Sani 2015), a SAT-Based Temporally Expressive Planner; YAHSP3 (Vidal 2014), which computes lookahead relaxed plans and uses them in state-space heuristic search; and Temporal FD (TFD) (Eyerich, Mattmüller, and Röger 2012), that uses context-enhanced additive heuristic over a temporal search space.

In contrast, MA-LAMA opens a new way to study temporal domains, as we aim to deal with the temporal complexity of concurrent actions search spaces with only MA techniques, by making use of the necessary temporal techniques only to maintain temporal and numeric soundness.

## Background

Following PDDL2.1 semantics (Fox and Long 2003), we define the input for our planning algorithm as:

### Definition 1. Temporal Planning Tasks

*A temporal planning task is defined as $\Lambda = \langle \rho, \vartheta, O_{inst}, O_{dur}, s_0, g \rangle$ where:*

- *$\rho$ is a set of atomic propositional facts,*
- *$\vartheta$ is a set of real-valued numeric fluents,*
- *$O_{inst}$ is a set of grounded instantaneous actions,*
- *$O_{dur}$ is a set of grounded durative actions,*
- *$s_0$ is the initial state, and*
- *$g$ is the goal condition.*

Instantaneous actions, $a_{inst} \in O_{inst}$, and durative actions, $a_{dur} \in O_{dur}$, differ from each other in the fact that durative actions take time, $dur(a_{dur})$, to perform a state transition from $s$ to $s'$. Instantaneous actions preconditions, $pre(a_{inst})$, and effects, $eff(a_{inst})$ are expanded to start conditions, $startCond(a_{dur})$, end conditions, $endCond(a_{dur})$, over all $dur(a_{dur})$ conditions, $inv(a_{dur})$, start effects, $startEff(a_{dur})$, end effects $endEff(a_{dur})$ and numeric effects, $contEff(a_{dur})$.

Start and end endpoints of $a_{dur}$, $a_\vdash$ and $a_\dashv$, can be encoded as instantaneous actions, with $pre(a_\vdash) = startCond(a_{dur})$, $eff(a_\vdash) = startEff(a_{dur})$, $pre(a_\dashv) = endCond(a_{dur})$, and $eff(a_\dashv) = endEff(a_{dur})$. This decomposition produces *snap actions*, allowing planners to reason with concurrent operators, as it allows them to overlap. The invariant condition, $inv(adur)$, for a durative action $(a_{dur})$ must be maintained throughout the open interval between $a_\vdash$ and $a_\dashv$.

For the numeric effects, we impose the same restrictions as the COLIN family of planners: the contribution of any durative action to the rate of change of each numeric fluent, $\upsilon \in \vartheta$, remains constant, so the rate of change of a certain variable, $\delta\upsilon$, only is modified when the *snap actions* are applied.

Additionally, a metric, $M$, can be defined to determine the quality of a plan, and it would be the planner's duty to find

which plan achieves a higher optimization of that metric. Different planners support a wide range of metric formulations, as the *total-cost* implementation in LAMA (Richter and Westphal 2010) and time-dependent continuous costs in OPTIC. In our case, we define $M$ as a set of weighted numeric variables, $\{w_1 * v_1, w_2 * v_2, ..., w_n * v_n\}$ where $v_n \in \vartheta$ and $w_n$ is a real number. $v$ can also be the duration of the plan, the *total-time*.

From this input, MA-LAMA translates it and utilises internally the multi-valued planning tasks (MPTs) representation, also referred as $SAS^+$ planning problems (Bäckström and Nebel 1995). For ease of presentation, we consider a simplified version of MPTs that omits axioms and with conditional effects compiled away.

**Definition 2. Multi-valued planning tasks (MPTs)**
*A multi-valued planning task (MPT) is a 4-tuple $\langle V, I, G, A \rangle$ where:*

- *$V$ is a finite set of multi-valued variables $v$,*

- *$I$ is the initial state for $V$,*

- *$G$ is the goal condition and a partial state of $V$, and*

- *$A$ is a set of instantaneous actions.*

The key difference between the two planning task representations is that variables in an MPT are multi-valued, rather than standard booleans in PDDL2.1. However, for our work we need to extend the MPT definition to fully encompass all the temporal task components, thus, we include the set of multi-valued numeric variables, $N$, and the metric, $M$, resulting in our extended MPT (eMPT) definition:

**Definition 3. Extended Multi-valued planning tasks (eMPTs)**
*An extended multi-valued planning task (MPT) is a 6-tuple $\langle V, I, G, A, N, M \rangle$ where:*

- *$V$, $I$, $G$, $A$ are defined as in the MPT Definition.*

- *$N$, a finite set of multi-valued numeric variables, $n$, each defined by a real numeric value and a finite set of exclusive states, and*

- *$M$, a metric to measure the plan quality, directly assigned from the temporal task.*

This extension allow us to deal with any numeric operation as an *effect* over $N$. We will expand on how to calculate the possible values for each numeric variable $n$ in the MA-LAMA details section.

The MPT representation also allows us to build the *casual graph* (CG), which represents dependencies among variables according to the available actions, and is the root for the MA task decomposition techniques.

**Definition 4. Casual Graphs**
*For a MPT $\Phi$ with a set of variables $V$, the Casual Graph of $\Phi$, $CG(\Phi)$ is the directed graph with a set of vertex $V$ that contains an arc $(v, v')$ iff $v \neq v'$ and there exists an action that can affect the value of $v'$, requiring a precondition that specifies the value of $v$.*

## MA-LAMA Overview

MA-LAMA is a satisfying temporal planner that utilizes MA task decomposition and required cooperation techniques to deal with the temporal complexity of concurrent actions search spaces. Additionally, it is designed to deliver fast and highly optimized solutions for MAP tasks.

Let us provide a comprehensive overview of the internal functioning of MA-LAMA, shown in Figure 1. The first step involves a translation of the temporal task, where the durative actions are transformed to *snap actions*, adding the same control predicates between $a_\vdash$ and $a_\dashv$ actions as the COLIN planner (Coles et al. 2012). Then, the *snap* temporal task is encoded into an eMPT and two MA algorithms take place:

- *Agent Decomposition (AD)*: following the ideas of ADP (Crosby, Rovatsos, and Petrick 2013), MA-LAMA decomposes the eMPT in terms of mostly independent entities, called agents.

- *Goal Categorization and Assignment (GCA)*: *coordination points* are computed following the principles of required cooperation (Zhang, Sreedharan, and Kambhampati 2016), and the task goals are categorized into *cooperation* and *coordination* subsets. Then, the goals are assigned to agents based on metric estimations, creating individual eMPTs to be solved subsequently and in groups, in what we call *Search Phases*.

Our objective with these two algorithms is to be able to strive for near optimal Search Phases solving, and, although we are aware that our decomposition results in a non-complete search space, we assume this compromise to achieve efficiency.

In each Search Phase, several eMPTs are solved, taking as an input the *temporal constraints* imposed by the previous agent eMPT and, similarly, Search Phases inherit the initial state from the previously solved one solution.

For each eMPT, we launch a modified version of the LAMA planner, with numeric, temporal and constraints frameworks built on top. Each eMPT is solved by a *WA\** search using two classical heuristics: $h_{FF}$ (Hoffmann and Nebel 2001) and $h_{Land}$ (Hoffmann, Porteous, and Sebastia 2004) (Sebastia, Onaindia, and Marzal 2006).

All eMPTs in each Search Phase generate a partial plan based on *snap actions*, so, after all phases are solved, we need to launch a Unify module that translates them to the temporal paradigm and assembles them, checking their temporal soundness and producing the full temporal plan.

Although privacy discussion is not a main topic for this work, we can safely state that MA-LAMA preserves *weak* privacy between the agents found by the AD, but not if a decomposition is not found, as MA-LAMA continues and solves the single-agent complete eMPT in this case.

## MA-LAMA insight

MA-LAMA makes use of the LAMA MPT representation, inherited from Fast Downward (FD) (Helmert 2006), which only needs to be modified to support the snap task numeric conditions, *numCond*, and any form of *contEff* during the temporal task instantiation and multi-valued variables computation through the invariant search.

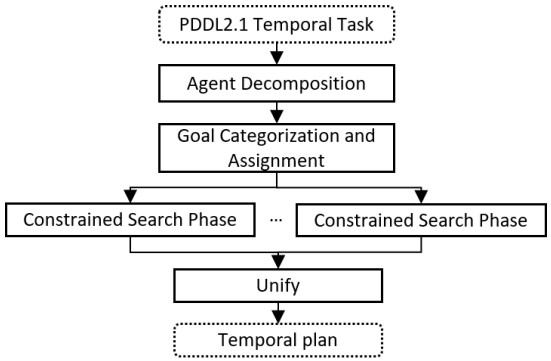

Figure 1: MA-LAMA general structure. The input is a PDDL2.1 temporal task that is decomposed to launch several Constrained Search Phases. All partial plans are unified at the end to produce a full temporal plan as output.

Additionally, as we have expanded the original MPTs definition to eMPTs, we need to determine the possible values for the multi-valued numeric variables, $n \in N$. For each numeric variable $n$, the set of states, $\epsilon$ is obtained by: $\forall a \in A \rightarrow \epsilon = \epsilon \cup \{n \in N : n \in contEff(a)\}$, plus the *undefined state*, $u$. *contEff*($a$) and *numCond*($a$) are expanded so that all numeric operations that can be solved before the search are solved. Thus, in our eMPTs and for each fluent numeric variable, we encode the current numeric value and last applied numeric effect.

Next, we will review in detail each MA-LAMA execution module, starting with Agent Decomposition and Goal Categorization and Assignment, and continue with Constrained Search and Unify, as shown in Figure 1

## 1- Agent Decomposition

Most variable decompositions divide the domain variables between *private*, variables (that cannot be changed by and are not required by any other agent) and *public* (representing the environment in which the agents operate), $P$.

We borrow from Crosby et al. (2013) their *variable decomposition* definition. In short, a variable decomposition of an eMPT divides it in a set of variable groups, $\Omega = \{\Omega_1, \Omega_2, ..., \Omega_n\}$, and a set of public variables, $P$, that can be empty. In their work, they also define *Agent Variable Decompositions* (ADP); variable decompositions where there are no *joint actions* and no external actions that can affect any agent internal variables. We do not need to impose this restriction, as we will deal with *required cooperation* in a later stage.

Algorithm 1 gives a pseudo-code overview of the MA-LAMA Agent Decomposition (AD) module, which produces a set of eMPTs $\langle V, I, G, A, N, M \rangle$, where all components are set with the exception of $G$.

We consider three main stages in the AD module:

**1) Find Possible Agents** starts with the ADP basis: removing 2-way cycles from the $CG$ and take resulting root nodes that still have one successor left as possible agents. Then, the algorithms differ, as we aim for different objectives with the decompositions: minimize local agent search space for

---

**Algorithm 1: Agent Decomposition (AD)**

**Input**: $eMPT \langle V, I, G, A, N, M \rangle$
**Output**: Agent Decomposed Task set $\Phi = \{\Phi_1, \Phi_2, ..., \Phi_n\}$
1: **Find Possible Agents**
2: $CG$ generation
3: $\Omega \leftarrow \{\nu \in V : \nu$ root node of $CG \backslash 2$ way cycles$\}$
4: $\Omega \leftarrow$ AssembleAgents($\Omega$)
5: **Extend Private Agent Sets**
6: **repeat**
7:     **for** $\Omega_n \in \Omega$ **do**
8:         $\Omega_n \leftarrow \Omega_n \cup \{v \in V : v$ only successor of $\cup \Omega_n\}$
9:     **end for**
10: **until** $\Omega$ can no longer be refined
11: $A_n \leftarrow \{a \in A, v \in \Omega_n : \exists v \in \cup eff(a) \cup \cup pre(a)\}$
12: **Extend Public Agent Sets**
13: **repeat**
14:     **for** $\Omega_n \in \Omega$ **do**
15:         $\Omega_n \leftarrow \Omega_n \cup \{v \in V : v$ is connected with $\cup \Omega_n$ in the $CG \wedge v$ not yet assigned$\}$
16:     **end for**
17: **until** every $v \in V$ is assigned
18: $A_n \leftarrow \{a \in A : \cup eff(a) \cup \cup pre(a) \in \Omega_n \wedge a$ not assigned in previous step$\}$
19: $N_n \leftarrow \{n \in N : n \in \cup contEff(\{a \in A_n\}) \cup \cup numPre(\{a \in A_n\})\}$
20: $I_n \leftarrow \{v \in I : v \in V_n\}$
21: $M_n \leftarrow \{(w * n) \in M : n \in N_n\}$
22: **return** $\{\Omega_n, I_n, \emptyset, A_n, N_n, M_n\}$

---

MA-LAMA, and minimize mandatory agent coordination in ADP.

We run an *Assemble Agents* step before expanding all agent sets, so that possible agents are more refined and really coupled agents are merged. The full assembly step can be summarized in one rule:

*For two possible agents $[v, v']$ in a root node set $\Omega$, $v$ and $v'$ are assembled if there is a path between $v$ and $v'$ in the $CG$.*

**2) Extend Private Agent Sets** Agents sets are then expanded so that every $v \in V$ that is only successor of an agent set, $\Omega_n$, is added to it. This process is repeated until all sets can no longer be refined, and a set of actions is added to each agent such that, for every action $a \in A$, it is added to the private set of actions of a certain agent, $A_n$, if there exists a variable $v$ that exists in *eff*($a$) or *pre*($a$).

This results in a *private* set of variables, $\Omega_n$ and actions $A_n$ for each agent; therefore, the rest of the variables and actions are assumed *public*, $P$.

**3) Extend Public Agent Sets** Agent variable sets are completed with all the variables that are reachable in the $CG$ for a certain agent, $n$: actions not yet assigned are added to $\Omega_n$ following the same past rule, but using the new complete variable set; numeric variables $N$ and metric, $M$ are added to $N_n$ and $M_n$ if they appear in the *conteff*($a$) or *numCond*($a$) for every $a \in A_n$; and the initial state $I$ and metric $M$ are decomposed and added to an agent $n$ if they appear in $V_n$.

---

**Algorithm 2: Goal Categorization and Assignment (GCA)**

**Input**: Agent Task Set, $\Phi = \{\Phi_1, \Phi_2, ..., \Phi_n\}$, and goals, $G$
**Output**: Cooperation and Coordination Search Phases, $\sigma = \{\sigma_1, \sigma_2, ..., \sigma_n\}$

1: **Coordination Points Variables**
2: $CoorP \leftarrow \emptyset$ (Coordination Points)
3: **for** $\{\Phi_n, \Phi_m\} \in \binom{\Phi}{2}$ **do**
4: $\quad CoorP \leftarrow CoorP \cup \{v \in P : \exists (a_n \in A_n, a_m \in A_m) : v \in pre(a_n) \wedge v \in \text{eff}(a_m) \}$
5: **end for**
6: **Single Goal Relaxed Plans Obtention**
7: $G_{coop}, G_{coord} \leftarrow \emptyset$ (Coord and Coop goal sets)
8: **for** $g \in G$ **do**
9: $\quad Sol_g \leftarrow \emptyset$ (relaxed solutions set)
10: $\quad$ **for** $\Phi \in$ **do**
11: $\quad\quad Sol_g \leftarrow Sol_g \cup$ **relaxedSearch**$(\Phi_n, g)$
12: $\quad$ **end for**
13: $\quad$ **if** $Sol_g \neq \emptyset$ **then**
14: $\quad\quad G_{coop} \leftarrow G_{coop} \cup g, Sol_g$
15: $\quad$ **else**
16: $\quad\quad Sol_g \leftarrow$ **relaxedSearch**$(\Phi, g)$
17: $\quad\quad Sol_g \leftarrow CoorP$ **in** $Sol_g$
18: $\quad\quad G_{coord} \leftarrow G_{coord} \cup g, Sol_g$
19: $\quad$ **end if**
20: **end for**
21: **Goal Assignment and Search Phases Creation**
22: $\sigma_{coop} \leftarrow G_{coop}, MinCostAssignment(\Phi, G_{coop})$
23: $\sigma_{coord} \leftarrow \emptyset$
24: **for** $g_{coord} \in G_{coord}$ **do**
25: $\quad \sigma_n \leftarrow g_{coord} \cup CoorP$ **in** $g_{coord}.Sol_g$, $\Phi_n$ **in** $g_{coord}.Sol_g$
26: $\quad \sigma_{coord} \leftarrow \sigma_{coord} \cup \sigma_n$
27: **end for**
28: **return** $\sigma_{coop} \cup \sigma_{coord}$

---

Our variable decomposition for an eMPT is both sound and complete as we meet all the same three criteria explained in Crosby et al. (2013), in Theorem 6.1.

Finally, the output of the algorithm is a set of tasks, $\Phi = \{\Phi_1, \Phi_2, ..., \Phi_n\}$, one for each agent and without any goal assigned. This set is the input for the Goal Categorization and Assignment stage.

## 2- Goal Categorization and Assignment

Algorithm 2 shows the Goal Categorization and Assignment (GCA). The objective is to further exploit the MA nature of the domains by studying how the goals from the original temporal task, $G$, can be assigned to the agent eMPT set, $\Phi$, to achieve optimized solutions.

The basis of the GCA algorithm is the Required Cooperation (RC) Analysis (Zhang, Sreedharan, and Kambhampati 2016), in which they formally describe the possible agent interactions within an eMPT. From now on, we will refer to goals that require Type-1 RC (Heterogeneous Agents) or Type-2 RC (Homogeneous Agents) Casual Loops interactions as *coordination* goals, and goals that require Type-2 RC Traversability interactions, or no RC at all, as *coopera-*

*tion* goals. The output of the GCA is a set of *Search Phases*, each one aiming to solve a goal subset of *coordination* or *cooperation* $\{g\} \in G$.

We consider three main stages for the GCA module:
**1) Coordination Points Variables** *Coordination points* are certain points in an agent plan where it possibly influences or is influenced by other agents. Following this idea, we obtain variables that may be *coordination points* in our $\Phi$ by extracting from $P$ all variables that are both a precondition in one agent actions and an effect in other agent actions.
**2) Single Goal Relaxed Plans Obtention** Then, we launch a relaxed (ignore delete effects and ignore *numCond()*) search for each goal $g \in G$ and agent eMPT $\Phi_n \in \Phi$ and calculate a metric value for the relaxed solution, computing numeric variables limits and using the worst-case scenario for fluents in *contEff()*.

If a solution is found for any of $\Phi_n$, then the goal is considered a *cooperation* goal, and our aim is then changed to find the most optimized relaxed solution through iterative relaxed searches and for each agent that can achieve it.

If a solution is not found, $g$ requires *coordination*, and a relaxed search is launched for the same $g$ with the full eMPT, trying to minimize the number of used agents in the relaxed solution. The values for variables selected as possible *coordination points* are stored for later use in the Search Steps creation. The optimization of all relaxed search processes take place for a configurable amount of time or until the complete search space has been explored.
**3) Goal Assignment and Search Phases Creation** We deal with *coordination* and *cooperation* goals in different ways:

- for *cooperation* goals, all goals are assigned in a way that minimizes the total sum of all relaxed metrics in a single Search Phase, $\sigma_1$, where agents involved, $\Phi_n \in \Phi$, are those with at least one goal assigned, and

- for *coordination* goals, one Search Phase is created for each, $\{\sigma_2, ..., \sigma_n\}$, where agents involved are those that appear in the relaxed plan, additionally, the *coordination points* for each agent are also assigned as goals.

We check if the assignments are valid from a numeric conditions perspective before continuing. If there are no relaxed solution for one of the agents in a Search Phase, the process is restarted assigning weights to each agent metric estimation, lowering the use of constrained agents.

In average, the time it takes for the GCA to find the best relaxed solution is in the order of milliseconds.

## 3- Constrained Search

The Constrained Search process receives as an input the full set of Search Phases, $\sigma = \{\sigma_1, \sigma_2, ..., \sigma_n\}$, each with a set of eMPT, $\Phi = \{\Phi_1, \Phi_2, ..., \Phi_n\}$. We launch a multi-heuristic constrained search over each eMPT, inheriting *temporal constraints*, $(\Phi_n, \upsilon, t, d)$, set by the public variables, $P$, between each agent search, so that interactions between agents are only considered when necessary.

**Definition 5. Temporal Constraints**
*A Temporal Constraint, $(\Phi_n, \upsilon, t, d)$, states that the agent, $\Phi$, sets a certain value, $\upsilon$, at the moment, $t$, for the a certain duration, $d$.*

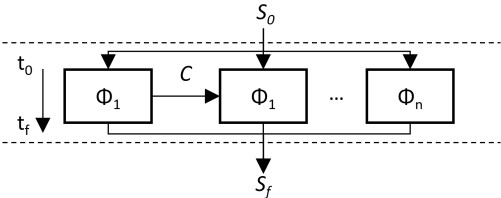

Figure 2: A Cooperation Search Phase: each agent, $\Phi_n$, produces *temporal constraints* for the next. All agents share initial state, $S_0$, and produce a combined final state, $S_f$. The phase makespan, $t$, is the longest agent plan makespan.

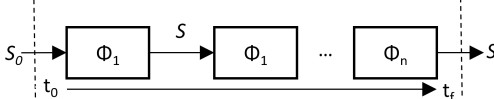

Figure 3: Coordination Search Phase structure, where each agent, $\Phi_n$, produces the initial state, $S$, for the next. The final state, $S_f$ is produced by the last agent. The makespan of the phase, $t$, is the sum of all agent local plans.

We will now divide the search description in two: first, each individual search characteristics are explained, and then, details on solving whole Search Phases are provided.

**1) Search details** For each eMPT, We launch a *WA\** forward total-order search with two classical heuristics, $h_{FF}$ (Hoffmann and Nebel 2001), Cost-Sensitive FF/add variant, and $h_{Land}$ (Hoffmann, Porteous, and Sebastia 2004) (Sebastia, Onaindia, and Marzal 2006). This choice is based on the fact that most MA temporal domains revolve their temporal complexity over the concepts of *cooperation* and *coordination*, so the eMPTs that we solve at this point tend to not require complex necessary simultaneity.

During search, all states are evaluated with respect to both heuristics, and, when choosing which state to expand, the search algorithm alternates between both based on numerical priorities. Inherited from the LAMA planner, we also make use of *preferred-operators*, which represent operators that are estimated to be useful in a given state.

A temporal framework is introduced to deal with local concurrency, incorporating constraints among snap-actions to guarantee that the preconditions for the new actions are satisfied in the frontier state, as well as keep track of the makespan and running actions start-end times for each state.

We follow the same principles with the numeric framework, including the necessary mechanisms to be able to deal with continuous numeric operations and numeric preconditions. Note that our only aim with these frameworks is to guarantee temporal and numeric soundness, and that we do not reason with time or incorporate other numeric solutions as *LPs* for the sake of fast search graphs exploration.

**2) Solving Search Phases** All Search Phases need a *temporal constraints* system for two main purposes:

- for *cooperation* goals, to assure that restrictions over variables $v \in P$ are preserved, and

- for *coordination* goals, to synchronize the agents around the *coordination points*.

In practice, *temporal constraints* are used as conditions in *cooperation* Search Phases (see Figure 2), computing them as *inv()* $= v$, starting at time, $t$, for the duration, $d$. When a *cooperation* agent finds a solution, it also computes a set of Temporal Constraints, containing all times a variable $v \in P$ value was required or changed. Consecutive agents use this list as limitations to their own local search graphs, and adding their own restrictions when they find a solution.

We solve *coordination* Search Phases, Figure 3, following the agent order dictated by the *coordination points*, obtaining the *coordination* goal at the end. Each agent inherits from the last a set of *temporal constraints* representing the already obtained subgoals and end states, which serve as the initial state for each local search.

Metric optimization in each Search Phase type is achieved differently. First, a *cooperation* Search Phase will be as optimized as its individual eMPTs are. Since several goals are achieved per agent, the ones with more goals are prioritized, so they solve their eMPTs less restricted by *temporal constraints*. *Coordination* Search Phases eMPTs generally only solve one goal, so they are not as promising in terms of numeric optimization. On the other hand, temporal concurrency can still be improved, and it is handled in the next step.

## 4- Unify

The final step in MA-LAMA execution consists of the unification of all partial plans for each Search Phase to obtain a full temporal plan.

The unification process of each agent in a Search Phase, $\Phi_n \in \sigma_j$, *snap* partial plan, is simple, as we have already dealt with concurrency and constraints in all cases but in between *coordination* Search Phases. The partial plans for each Search Phase are obtained by assembling each $\Phi_n$ partial plan in a concurrent manner for *cooperation* Search Phases, and consecutively in *coordination* Search Phases.

In order to combine all partial plans, we first check for each *coordination* Search Phase pair, $\sigma_n, \sigma_m$, and the variables from $P$ that are affected in their respective *snap* partial plans, $P_n, P_m$. If $P_n \cap P_m = \emptyset$, then both $\sigma_n$ and $\sigma_m$ are added to the complete temporal plan in a concurrent manner.

Finally, all remaining partial plans are combined consecutively. During this process, we also calculate the total cost of the final plan; check that the temporal, numeric and logic constraints soundness is maintained; and change the *snap* actions plan paradigm to temporal.

## Experimentation

Our experimentation is divided in two sections. First, we study the coverage results of MA-LAMA in temporal and non temporal domains against other classical and temporal solvers, in order to check if the AD and GCA MA algorithms are suitable to deal with a wide range of MA scenarios. And, second, we analyse MA-LAMA plan quality performance for increasingly difficult temporal problems, and against other state-of-the-art temporal planners.

| CoDMAP Domains | MA-L | CMAP | ADP | | | Goal Types |
|---|---|---|---|---|---|---|
| Blocksworld | **20** | 19 (1) | **20** | | | - |
| Depot | **17 (3)** | **17 (3)** | 16(4) | | | - |
| DriverLog | **20** | 19 (1) | **20** | | | Coop |
| Elevators | **20** | 18 (2) | **20** | | | Coop&Coor |
| Logistics | **20** | 19 (1) | **20** | | | Coop&Coor |
| Rovers | **20** | **20** | **20** | | | Coop |
| Satellites | **20** | **20** | **20** | | | Coop |
| Sokoban | 14 (6) | 13 (7) | **17(3)** | | | - |
| Taxi | **20** | **20** | **20** | | | Coop |
| Wireless | 6 (14) | 5 (15) | 9(11) | | | - |
| Woodworking | 16 (4) | 15 (5) | **20** | | | - |
| Zenotravel | **20** | 19 (1) | **20** | | | Coop |
| **Total** | 213 | 204 | **222** | | | Coop |
| Domains (origin) | MA-L | TFLAP | OPTIC | TFD | POPC | Goal Types |
| Rovers (IPC) | **11** | 1(10) | 8 (3) | 10 (1) | 6 (5) | Coop |
| Satellite (IPC) | **11** | **11** | 7 (4) | 5 (6) | 0 (11) | Coop |
| Zenotravel (IPC) | **16** | 14 (2) | 8 (8) | **16** | 7 (9) | Coop |
| Logistics (CodMAP) | **20** | **20** | **20** | **20** | 0 (20) | Coop&Coor |
| Taxi (CodMAP) | **20** | 18 (2) | **20** | **20** | **20** | Coop |
| **Total** | **78** | 54 | 63 | 71 | 33 | Coop |

Table 1: Coverage results (not solved domains in parenthesis) for CoDMAP non-temporal (up) and CoDMAP and IPC temporal (down) MA domains. MA-LAMA goal types detected are shown for domains with valid decompositions. All executions are limited to 10 minutes and 4GB of RAM.

## Coverage for classical and temporal tasks

We first check the coverage of MA-LAMA in classical and temporal domains, results can be seen in Table 1.

For classical planning, we analyse over the CoDMAP domains and against the ADP-legacy and CMAP-q planners. These are the winners for the competition coverage and quality tracks, and both share internal functionalities with MA-LAMA: ADP shares the root of the agent decomposition, launching FF if the decomposition is not valid; and CMAP-q, that obfuscates the full domain information internally and makes use of the LAMA planner during search.

We obtain a similar coverage performance to ADP when an agent decomposition is found, and to CMAP-q if the decomposition is invalid (generally, when the AD delivers a decomposition that does not match the competition original domain). Compared with ADP, the slightly worse coverage performance of MA-LAMA is explained by this, as we solve less domains when no decomposition is found. Overall, MA-LAMA achieves a coverage of 88,75%, compared to 92,5% from ADP-legacy and 85% from CMAP-q.

For MA temporal domains, we compare against other state-of-the-art temporal planners, already mentioned in the related work section: OPTIC, TFLAP, TFD and POPCORN. We exclude Yahsp3, as its single-thread version is outperformed by TFD and does not work with metrics in our tests, and both TemPorAl and CP4TP, as they are portfolios and the comparisons would not be significant. The domains we chose are Rovers, Satellites and Zenotravel from IPC, and adapted Taxis and Logistics from CodMAP to make them temporal, as Logistics presents required cooperation and Taxis allow us to check a less complex scenario.

In this case, MA-LAMA outperforms all planners, specially for IPC domains, where only TFLAP in Satellites and TFD in Zenotravel are able to match MA-LAMA. Regarding the agent decompositions, we obtain the ones a human operator would set: planes in Zenotravel, rovers in Rovers, satel-

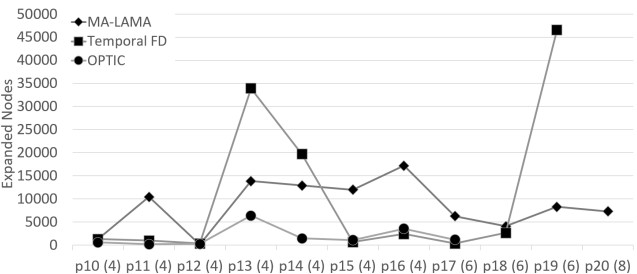

Figure 4: Expanded Nodes for each problem ($n^o$ of agents) in Rovers IPC domain.

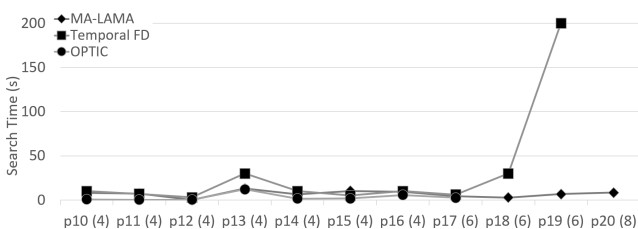

Figure 5: Search Time in seconds for each problem ($n^o$ of agents) in Rovers IPC domain.

lites in Satellites, planes and trucks in Logistics and taxis in Taxis. Additionally, only Logistics contained *coordination* goals, but this did not affect coverage performance. With these results, we can conclude that the AD and GCA algorithms are suitable to deal with MA temporal domains.

The coverage results show that MA-LAMA is able to solve a wide range of problems in MA classical domains, and improves state-of-the-art coverage performance for the most complex MA temporal domains.

## Plan quality performance

In this section, we will study the plan quality performance in MA temporal domains, the main focus of our planner. For the reasons we outlined in the previous point, we launch the same five planners: OPTIC, TFLAP, TFD, POPCORN and MA-LAMA. Results are detailed in Table 2. For all domains, we consider problems once the MA nature becomes relevant.

MA-LAMA dominates in Rovers (weighted battery) and Zenotravel (weighted fuel + makespan) domains, delivering better quality plans in all instances for Rovers and almost all for Zenotravel, where TFLAP also wins in some problems. These two domains are launched with metrics that are not completely coupled with the makespan of the plan, and the two best performing planners (TFLAP only in Zenotravel) make use of classical heuristics and do not reason with time, meaning that this can be an advantage when optimizing metrics not coupled with the makespan.

For Satellites (makespan), result are mixed, MA-LAMA gives better solutions in four problems, and OPTIC and TFD in three. It is clear that planners that reason with time perform better with the makespan metric, but, as instances get more complex, MA-LAMA deals better with them and is able to deliver good solutions in these last problems. This

| Domain (metric) | Problem | 6 | 7 | 8 | 9 | 10 | 11 | 12 | 13 | 14 | 15 | 16 | 17 | 18 | 19 | 20 |
|---|---|---|---|---|---|---|---|---|---|---|---|---|---|---|---|---|
| Rovers (*w* battery) | MA-LAMA | | | | | **48.6** | **18.99** | **88.99** | **155.8** | **79.8** | **102.3** | **77.0** | **135.3** | **33.2** | **313.4** | **388.6** |
| | POPCORN | | | | | 130.2 | 37.2 | 93.4 | - | - | 156.8 | 184.3 | - | 136.6 | - | - |
| | TFD | | | | | 101.1 | 47.2 | 100 | 172.1 | 94.1 | 215.2 | 100.2 | 214.6 | 224.4 | 346.0 | - |
| | TFLAP | | | | | - | - | - | - | - | 137.3 | - | - | - | - | - |
| | OPTIC | | | | | 99.9 | 38.8 | 96.5 | 215.6 | 143.8 | 184.7 | 198.7 | 217.4 | - | - | - |
| Satellites (makespan) | MA-LAMA | | | | | 134.50 | 205.90 | 230.25 | 132.89 | 129.12 | **192.16** | **120.18** | 93.54 | 153.60 | **247.92** | **555.90** |
| | TFD | | | | | - | - | 262.45 | **104.05** | **87.71** | - | - | - | **93.92** | 283.43 | - |
| | TFLAP | | | | | 217.57 | 479.33 | 385.36 | 634.91 | 361.01 | 462.91 | 370.22 | 385.51 | 296.31 | 509.53 | 948.83 |
| | OPTIC | | | | | **115.74** | **150.91** | **171.79** | 149.10 | 108.63 | - | - | **83.74** | 100.63 | - | - |
| Zenotravel (makespan + *w* fuel) | MA-LAMA | **19.83** | **54.47** | **45.95** | **35.92** | 237.72 | 76.7 | 103.17 | 123.7 | **268.26** | 182.61 | **210.3** | **324.42** | 154.64 | **373.43** | **398.15** |
| | POPCORN | 124.7 | 127.01 | 366.1 | 719.69 | - | 313.74 | 213.49 | - | - | - | - | - | - | - | - |
| | TFD | 39.77 | 111.19 | 236.7 | 651.96 | 268.73 | 208.24 | 146.42 | 194.72 | 1464.59 | 242.81 | 574.42 | 1756.12 | 880.85 | 1674.44 | 3301.31 |
| | TFLAP | 24.99 | 82.75 | 76.38 | 213.95 | **196.13** | 81.75 | **81.94** | **92.59** | 390.97 | **133.04** | 303.66 | 504.84 | 414.58 | - | - |
| | OPTIC | 38.66 | 66.93 | 102.17 | 131.03 | 283.3 | - | 400.28 | 130.07 | 291.74 | - | - | - | - | - | - |
| Logistics (*w* fuel) | MA-LAMA | 169.0 | **131.0** | 114.0 | **177.0** | 168.0 | 192.0 | **247.0** | 188.0 | 263.0 | **231.9** | **210.0** | **170.5** | 308.4 | 266.2 | **197.0** |
| | TFD | 183.0 | 217.0 | **104.0** | **177.0** | 168.0 | 304.0 | **247.0** | 168.0 | 521.0 | 434.4 | 332.4 | 216.2 | 290.0 | **221.0** | 235.2 |
| | TFLAP | **159.0** | **131.0** | 114.0 | **177.0** | **158.0** | 172.0 | **247.0** | **158.0** | **239.0** | 287.0 | 244.0 | 217.0 | **264.8** | 311.0 | 256.0 |
| | OPTIC | 183.0 | **131.0** | 114.0 | 187.0 | **158.0** | 206.0 | **247.0** | 178.0 | 253.0 | 272.4 | 213.9 | 192.8 | 318.0 | 313.2 | 229.2 |
| Taxi (*w* dist) | MA-LAMA | **5.6** | **4.6** | 5.1 | 7.1 | **4.6** | **4.6** | **6.5** | **7.5** | **17.5** | **18.9** | 27.3 | 23.4 | **19.8** | **28.3** | **31.2** |
| | POPCORN | 21.5 | 18.9 | 8.1 | 55 | 8.1 | 36.8 | 28.6 | 54.7 | 53.5 | 68.2 | 73.7 | 66.1 | 65.8 | 91.3 | 73.8 |
| | TFD | 16.8 | **4.6** | 7.2 | 14.2 | **4.6** | **4.6** | 11.8 | 15.3 | 53.4 | 34.5 | 43.5 | 24.3 | 24.3 | 61 | 55.4 |
| | TFLAP | **5.6** | **4.6** | 5.1 | **6.6** | **4.6** | **4.6** | **6.5** | **7.5** | **17.5** | **18.9** | **18.9** | **19.8** | **19.8** | - | - |
| | OPTIC | **5.6** | **4.6** | **4.6** | 21 | **4.6** | **4.6** | 29.1 | 45.7 | 20.1 | 22 | 58.8 | 35 | 69.8 | 75 | 73.5 |

Table 2: Quality of plans on IPC benchmarks (Rovers, Satellites and Zenotravel), and temporal CodMAP domains (Logistics and Taxi), limited to 10 minutes and 4GB of RAM. Smaller is better in all domains. Absence of a planner in a given domain indicates that it solved no problems. "*w*" means weighted.

trend is present in the domains.

The CodMAP domains are significantly less complex than the IPC ones, as several planners find optimal (or near optimal) solutions for several instances. In these cases, MA-LAMA depends on the GCA algorithm to deliver the best solution, as a non-optimal goal assignment results in a non-optimal final solution, which happens in several instances of Taxis (weighted distance) and Logistics (weighted fuel). Similarly to the previous case, as problems incorporate more agents and variables, all planners begin to struggle to optimize the solutions, and the MA nature of MA-LAMA proves to be an advantage in these cases for both domains.

Logistics has been incorporated as the case where *coordination* goals cannot be avoided, and MA-LAMA is able to find the optimal solution only in some easier domains, showing good performance in the latest ones. This proves that the GCA is only suitable for scenarios where finding optimal solutions in a short time span is not feasible, as it looses too much information in the estimation process for lighter ones.

Lastly, we want to take a look on the search time and expanded nodes for each planner in the Rovers domain, shown in Figure 4 and Figure 5. POPCORN and TFLAP are excluded since they do not solve the majority of the problems.

For OPTIC, and more notably for TFD, the increase in expanded nodes can be tied to an increase in search time, and increasing the number of agents causes them to struggle to find good, if any, solution. MA-LAMA does not reproduce these behaviors, expanded nodes and search time remain stable through the whole domain, with peaks in expanded nodes when the individual agent eMPTs are harder, as in *p16*. Additionally, the increase in the number of agents does not translate in worse performance, as the AD and GCA algorithms remove much of the temporal complexity that this certain domain presents.

To conclude, experiments infer that the MA techniques in MA-LAMA perform suitable decompositions for MA temporal domains, and that MA-LAMA delivers better plan quality performance than other state-of-the-art temporal solvers in the most complex problems for all tested domains.

## Future work and limitations

MA-LAMA does not reason with temporal or numeric information during search, so, in domains where a decomposition is not found, it is expected to underperform other planners. Other techniques, as symmetry based decompositions or search time reasoning could be incorporated for theses cases. Additionally, the classical heuristics used can run into plateaus during search due to the lack of temporal reasoning.

The AD algorithm can be improved to cover a wider range of MA domains, as currently there are several domains from CodMAP that remain hard to decompose automatically into logical sets of variables.

Finally, our experimentation shows that MA-LAMA improved plan quality performance over multi-agent temporal domains over the state-of-the-art, but it is expected to be dominated in other types of scenarios. A future line of work may consider including MA-LAMA and other planners to build a portfolio, launching only selected planners based on a domain structure analysis performed before the search.

## Conclusions

In this paper, we presented MA-LAMA, a satisfying temporal MA planner that utilises MA techniques to deal with concurrent action search spaces. The AD and GCA algorithms decompose the temporal MAP tasks in Search Phases in a way that reduces temporal complexity and that is suitable for metric optimization.

Our experimentation shows that MA-LAMA outperforms other state-of-the-art temporal planners in terms of coverage and plan quality in MA temporal domains, especially in the most complex domain instances.

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
