# OpenReview forum: "Multi-Agent Temporal Task Solving and Plan Optimization"
_icaps-conference.org/ICAPS/2024/Conference — ICAPS 2024_

### Official Review · Reviewer_5VEp · 2024-01-19

**Significance And Importance:** 3
**Soundness:** 4
**Novelty:** 3
**Clarity:** 4
**Overall Evaluation:** 2
**Confidence:** 3

**Weaknesses:**

2: No major or minor weaknesses.

**Contributions Of The Paper:**

The paper presents a multi-agent temporal planning approach. Integrating different techniquest, an algorithm is presented to deal with multi-agent temporal planning problems, to guarantee a good coverage of planning problems (considering some well known domains used in IPC) and also exhibit good performance.

**Ethical Considerations:**

(5) Excellent: The paper comprehensively addresses all of the applicable ethical considerations

**Nomination For Best Paper:**

No

**Questions For Authors:**

is it possible to assess the impact of agent decomposition decisions with respect to goal assignement?

Is it possible to evaluate whether these sequential decisions can actually influence the quality of the generated solutions?

**Reproducibility:**

3: Authors describe the implementation and domains in sufficient detail.

**Strengths Of The Paper:**

The paper provides an algorithm for multi-agent temporal planning leveraging task/goal assignment over multiple agent in an effective way.

**Weaknesses Of The Paper:**

The paper seems to me nice. A weak point can be a negative impact of sequencing solving decisions (Agent decomposition and Goal Assignment steps). Namely, the application in sequence of decisions taken on agent selection/goal assignments may prevent to find other (better) valid solutions.

---

> ### Author Rebuttal · Authors · 2024-01-27
>
> Thank you for your review, we really appreciate your comments.
>
> Regarding the matter raised in the "Weaknesses" section, we recognize that sequencing decision-making processes might prevent our planner from discovering alternative, potentially superior solutions that could be identified through interleaved decisions. This was a deliberate consideration during the design phase. We would like to justify our decisions and answer to your questions accordingly.
>
> The Agent Decomposition algorithm generates a variable decomposition based on independently acting entities, identified via the causal graph. The key decision in this decomposition is defining the nature of an agent (root nodes in the causal graph with loops removed) and determining the extent to which agents are merged if they interact within the causal graph.
>
> Question 1: Decomposition influence the Goal Assignment as both processes make decisions based in agent interactions. When two or more agents exhibit required cooperation, they may be treated as a single entity in the Agent Decomposition algorithm. For example: in the Logistics domain, trucks and planes might be viewed as separate agents, needing coordination to achieve goals, or they could be assembled into a 'super-agent' that can both operate a plane and a group of trucks. Consequently, the decisions made during Agent Decomposition affect how many coordination goals are identified in the Goal Assignment, as assembled agents would be able to solve these goals without any other agent involvement, at the cost of more complex local tasks.
>
> Question 2: These decisions also impact plan quality. MA-LAMA handles cooperation and coordination goals differently, but it is not consistently feasible during our informed Goal Assignment to predict whether individual or assembled agents will yield superior solutions once local tasks are resolved and integrated. Therefore, it is not guaranteed that interleaving these processes and selecting an agent assembly degree based on cost estimations would consistently lead to higher-quality plans. Our findings indicate that superior plan quality is closely linked to near-optimal resolution of local tasks. As such, our strategy is to prioritize single agents during Agent Decomposition, avoid interleaving goal assignment, and manage coordination through designated coordination points.

---

### Official Review · Reviewer_bg5b · 2024-01-21

**Significance And Importance:** 2
**Soundness:** 3
**Novelty:** 3
**Clarity:** 3
**Overall Evaluation:** 1
**Confidence:** 3

**Weaknesses:**

0: Minor weaknesses requiring some work to be addressed for the paper to be accepted.

**Contributions Of The Paper:**

The paper presents MA-LAMA, an extension of the LAMA planner that takes into consideration the multi-agent features of temporal planning domains to create a planner that performs better than state-of-the-art temporal planners.

The paper has an extensive discussion of the literature related to the development of MA-LAMA. It describes its key algorithms which include the explanations for different steps of the algorithms. The experimentation indicates that MA-LAMA provides good performance, coverage,and quality in several temporal domains used in the CoDMAP and IPC collection.

**Ethical Considerations:**

(1) Not Applicable: The paper does not have any ethical considerations to address

**Nomination For Best Paper:**

No

**Questions For Authors:**

1. Do the authors have proofs for the correctness of the algorithms?
2. This might not be a question but I was wondering whether MA-LAMA was tested with non-temporal domains such as logistics (multiple agents); if it was, whether MA-LAMA performs better than LAMA?

**Reproducibility:**

3: Authors describe the implementation and domains in sufficient detail.

**Strengths Of The Paper:**

The resulting planner, MA-LAMA, performs well and yields good quality plans in several domains when it is compared with state-of-the-art temporal planners.

The authors did a good job on reviewing related work.

**Weaknesses Of The Paper:**

The proofs for the correctness of the algorithms are missing.

---

> ### Author Rebuttal · Authors · 2024-01-27
>
> Thank you for your review, we really appreciate your comments.
>
> Find here the correctness Sketch of Proof affirming the soundness and completeness of both algorithms. These will be also included in the revised paper.
>
> For the Agent Decomposition (AD), the aim is to prove that our algorithm and decomposition definition are aligned. For this, we rely on Crosby et al. 2013 Theorem 6.1 Sketch of Proof, as we share most of the definition with them. First, if variables identified during the "Find Possible Agents" stage also serve as root nodes in the causal graph, it is guaranteed that a decomposition will be produced. And second, if a decomposition exists, the "Find Possible Agents" stage will always identify a minimum of two root variables that will not be merged.
> However, our AD does not satisfy their remaining statement: "for any pair of sets that violate the Agent Variable Decomposition property, it can be shown that they will be merged". As stated in our paper "Agent Decomposition" section, we do not impose this "merge restriction" to our definition, therefore, we are aligned with our own decomposition definition and achieve soundness and completeness despite this.
>
> For the Goal Classification and Assignment, we demonstrate that, given any valid agent decomposition, a valid and solvable goal assignment will be generated. First, if no decomposition has been found, all goals will be assigned to the full task. Second, if a certain goal can be attained by the full task, it will be assigned to an agent that can also attain it, as our first relaxed-searches only make use of each single agent internal operators, or it will be marked as a Coordination Goal and will be assigned to a group of involved agents through a full task relaxed-search. Finally, it can be demonstrated that coordination points for Coordination Goals cover the remaining case from the AD, involving linked agent sets in the causal graph.
>
> About the second question, we tested MA-LAMA in a wide range of classical and temporal domains, including both versions of logistics, where MA-LAMA attains superior solutions compared to base LAMA if the complexity of the problem makes it not feasible to be completely explored within a reasonable time. However, it is important to acknowledge that non-temporal problems exhibit substantially lower complexity compared to their temporal counterparts and, therefore, MA-LAMA only achieves better solutions than LAMA for the most complex non-temporal scenarios.

---

### Official Review · Reviewer_3wjv · 2024-01-24

**Significance And Importance:** 2
**Soundness:** 4
**Novelty:** 3
**Clarity:** 3
**Overall Evaluation:** 1
**Confidence:** 3

**Weaknesses:**

0: Minor weaknesses requiring some work to be addressed for the paper to be accepted.

**Contributions Of The Paper:**

A new  a centralized, unthreated, satisfying, total-order, multi-agent temporal planning system for multi-agent applications
Proposed system outperforms existing planners in several classical and temporal domains

**Ethical Considerations:**

(5) Excellent: The paper comprehensively addresses all of the applicable ethical considerations

**Nomination For Best Paper:**

No

**Questions For Authors:**

- The categorization provided in the 3rd paragraph of the Intro (from  (Torreno et al. 2017) is not completely aligned with the definition given for the proposed system "a centralized, unthreaded, satisfying, total-order, temporal MAP system". For example wrt privacy preservation doesn't seem to be in that definition. What is the categorization according to (Torreno et al. 2017)?
- the work proposes/includes ways to decompose the problem and solve them through this decomposition-unify process. I am wonder what is if there a certain type of characteristic of problems in which this decomposition approach fails to produce good plans.
- OPTIC seems to provide good results in the experiments. Do you consider the performance of the proposed planner to be significantly better than the other planners. Table 2 shows good results for MA-LAMA - it seems that it is ~2x better in some cases. I missed addidional data related to run time for instance, maybe there is significant difference between the runtimes?

**Reproducibility:**

3: Authors describe the implementation and domains in sufficient detail.

**Strengths Of The Paper:**

Addresses a relevant class of problems: multi-agent planning
Proposes an interesting approach that combines ideas and concepts from different efforts/work.
Good link with existing work both on the MA planning and temporal planning, making it an interesting blend of techniques.
Good explanation of the different steps of the algorithm
Good performance results provided. Planner outperforms existing temporal planners in many cases. It is interesting to see how OPTIC has also reasonable results in many cases too.

**Weaknesses Of The Paper:**

- minor: grammar through the paper has to be revisited/checked (e.g. start of system enumeration in the 2nd paragraph of related work, or the use of the term based in -> based on, or "makes us of", etc)
- minor: experiment includes non-temporal problems. I don't believe this is adding much here is the goal is to address temporal problem.

---

> ### Author Rebuttal · Authors · 2024-01-27
>
> Thank you for your review, we really appreciate your comments.
>
> Grammar will be thoroughly reviewed and errors corrected for the paper revision.
>
> For the non-temporal problems contribution. Our focus is MA temporal tasks, however, we conduct classical experiments to show that our MA algorithms are suitable for general MA tasks. We tie them with the results from temporal domain coverage to illustrate that traditional MA techniques are competent in handling temporal domains, as showcased in our work. This approach also provides the only feasible benchmark for comparison with other MA planners. We will explain the objectives of the classical results more clearly in the paper.
>
> As per Torreño et al. 2017, MA-LAMA is characterized as a Factored Planner, Centralized, Unthreaded with Pre-Planning Coordination (coordination points) and Iterative Response Planning (among cooperation agents), with Internal Communication and Local Heuristic Search. It preserves Weak Privacy if a decomposition is found, otherwise, No Privacy. This information will be incorporated in the paper.
>
> In terms of the decomposition-unify processes, these often encounter problems in tightly-coupled domains (as IPC Rovers and Logistics), where constraints complicate the resolution of local tasks. Our approach with the MA algorithms is to achieve the highest level of decomposition feasible, to ensure that tasks within the Search Phases are solvable even under heavy constraints. Other drawback of these processes is the incomplete exploration of the original search tree, meaning that the attainment of optimized solutions is reliant on the MA-LAMA goal assignment, which has consistently yielded positive results.
>
> Concerning the last question, we assert that MA-LAMA results are significantly better. Our metrics and domains selection is designed to encompass all planners strengths and weaknesses. Yet, MA-LAMA excels in all scenarios, including domains typically favorable to other planners, such as makespan in Satellites for OPTIC and TFD.
> While OPTIC demonstrates good results, it encounters difficulties with metrics not coupled with makespan and, like other planners, struggles with managing concurrency with high agent counts. MA-LAMA outperforms in these aspects.
>
> Regarding runtime data, runtimes are low and comparable in less complex domains. However, in more complex ones, MA-LAMA's speed is superior, occasionally by orders of magnitude. This aspect will be elaborated further in the revised paper.

---

### Meta-Review · Area_Chair_V1ac · 2024-02-05

**Recommendation:** Accept (Oral)
**Confidence:** 5

**Metareview:**

The authors present a new algorithm for solving multi-agent temporal planning problems. The approach is sound and well established with respect to the state of the art. The empirical evaluation is well conducting and highlights important benefits in several well established benchmarks.

The rebuttal provides important clarifications, notably regarding proof sketches and the imposed sequencing of the decision-making, that should be integrated in the final manuscript.

**Ethical Considerations:**

(1) Not Applicable: The paper does not have any ethical considerations to address